# Validity of constructed-response situational judgment tests in training programs for the health professions: A systematic review and meta-analysis protocol

Sara Mortaz Hejri[1], Jordan L. Ho[2], Xuan Pan[1], Yoon Soo Park[3], Amir H. Sam[4], Haykaz Mangardich[1], Alexander MacIntosh[1] *

1 Acuity Insights, Toronto, Canada, 2 Deloitte, Toronto, Canada, 3 Massachusetts General Hospital, Harvard Medical School, Boston, Massachusetts, United States of America, 4 Imperial College School of Medicine, Imperial College London, London, United Kingdom

* amacintosh@acuityinsights.com

## Abstract

### Background

Situational judgments tests have been increasingly used to help training programs for the health professions incorporate professionalism attributes into their admissions process. While such tests have strong psychometric properties for testing professional attributes and are feasible to implement in high-volume, high-stakes selection, little is known about *constructed-response* situational judgment tests and their validity.

### Methods

We will conduct a systematic review of primary published or unpublished studies reporting on the association between scores on constructed-response situational judgment tests and scores on other tests that measure personal, interpersonal, or professional attributes in training programs for the health professions. In addition to searching electronic databases, we will contact academics and researchers and undertake backward and forward searching. Two reviewers will independently screen the papers and decide on their inclusion, first based on the titles and abstracts of all citations, and then according to the full texts. Data extraction will be done independently by two reviewers using a data extraction form to chart study details and key findings. Studies will be assessed for the risk of bias and quality by two reviewers using the "Quality In Prognosis Studies" tool. To synthesize evidence, we will test the statistical heterogeneity and conduct a psychometric meta-analysis using a random-effects model. If adequate data are available, we will explore whether the meta-analytic correlation varies across different subgroups (e.g., race, gender).

### Discussion

The findings of this study will inform best practices for admission and selection of applicants for training programs for the health professions and encourage further research on constructed-response situational judgment tests, in particular their validity.

**Data Availability Statement:** No datasets were generated or analysed during the current study. All

relevant data from this study will be made available upon study completion.

**Funding:** The authors received no specific funding for this work.

**Competing interests:** YSP, AHS, and JLH have no disclosures to declare. SMH, XP, HM, and AM disclose they are salaried employees of Altus Assessments which administers a situational judgment test called Casper. The authors receive no reimbursements, fees, or funding related to this study or its outcomes. This does not alter our adherence to PLOS ONE policies on sharing data and materials.

## Trial registration

The protocol for this systematic review has been registered in PROSPERO [CRD42022314561]. https://www.crd.york.ac.uk/prospero/display_record.php?ID=CRD42022314561.

## Introduction

Training programs for the health professions (e.g., medicine, nursing, physician assistants, physical and occupational therapy, etc.) have traditionally prioritized applicants' academic knowledge and skills, such as their grade point average (GPA) and Medical College Admission Test (MCAT) scores as indicators of their future success [1]. However, this emphasis on academic-based metrics has demonstrated a major limitation: curtailing the selection of applicants who are not just 'book-smart,' but also possess the professionalism and interpersonal-based attributes required for success as healthcare practitioners [2,3]. The newfound importance of evaluating personal and professional characteristics in admissions (e.g., empathy, integrity) has sparked a rise in holistic review by programs seeking valid and legally defensible assessment methods [4].

Situational judgments tests (SJTs) have been increasingly used to address the gap in programs' ability to incorporate attributes outside of academic metrics into their admissions process. SJTs are assessments that present applicants with a series of hypothetical scenarios and assess their responses to those situations, with the goal of evaluating how someone is likely to react or behave in a given setting [5]. SJTs are favorably received by applicants [6], are cost-effective and feasible to implement in high-volume, high-stakes assessments, and thus, can be easily used in the earlier stages of selection [7]. One meta-analysis showed that SJTs have an internal consistency coefficient that ranged from 0.43 to 0.94 [8]. However, it is often argued that lower internal consistency of SJTs is found in cases where the test has a heterogeneous structure representing many different factors [9,10]. Also, several studies have explored SJTs' validity—defined as the test's ability to differentiate among people on other variables that measure the same or conceptually relevant constructs, behaviours, and performances, and operationalized as the correlation between scores on SJTs and scores on relevant outcomes [11–15]. While some studies have found evidence to support the interpretation that SJT scores can explain unique variance in relevant outcomes not explained by other measures [11–13], more recent studies have reported varying levels of validity depending on the type of SJT and outcome. A recent systematic review has found a moderate level of validity for SJTs (pooled estimate of 0.32, p value < 0.0001) [14], and another systematic review found that the SJT subtest of The University Clinical Aptitude Test (UCAT) was a weak predictor of professional behaviour during medical school [15].

To help elucidate why the above-mentioned findings conflict, it is possible that there are moderating characteristics that influence the magnitude of SJT validity values. Past meta-analyses have shown that moderators such as test construction (e.g., consultation of subject matter experts, level of detail in scenarios), study design (e.g., concurrent vs. predictive), and response instructions (e.g., knowledge vs. behavioral tendency) can affect the validity of SJTs [9]. However, little attention has been directed toward understanding one potential moderator in particular: the role of SJTs' response format. It is unclear, for instance, how the validity of constructed-response SJTs (i.e., where applicants give free-text answers) compares to that of selected-response formats (i.e., multiple-choice). This distinction between the two SJT formats

is crucial to make. The results of previous studies have demonstrated that, while selected-response questions can give a false impression of students' competence, constructed-response exams have advantages in terms of reliability, validity, and distinction between high and low performers [16,17]. Specifically regarding SJTs, constructed-response formats are less susceptible to faking by applicants [7], and have increased ability to discriminate between applicants compared with selected-response questions [18]. These differences suggest that constructed-response SJTs may have greater validity than selected-response formats. With the widespread and growing use of constructed-response SJTs by training programs for the health professions [19–21], investigating the validity of this format has strong practical implications.

Another important aspect of the admission process is the impact of selection tools on the shape of diversity. When using SJTs, it is important to know the extent to which comparable inferences can be made across different demographic subgroups (e.g., gender, race, ethnicity, and socioeconomic status) [22,23]. Research has demonstrated, for instance, that subgroup differences were significantly smaller or reversed for constructed-response SJTs, not only in comparison to other selection measures such as GPA and the MCAT [21], but also when compared with selected-response SJTs [23]. Nonetheless, the magnitude of subgroup differences is not equivalent to comparable validity inferences across groups. The present research will therefore seek to contribute new knowledge by directly examining the validity of constructed-response SJTs across demographic groups and identity characteristics.

To address the above-mentioned literature gaps, the present systematic review and meta-analysis will address the following primary question:

- What is the magnitude of the relationship between constructed-response SJT scores and other measures of personal, interpersonal, and professional attributes assessed either concurrently or in the future in training programs for the health professions?

Depending on the data availability, the present systematic review will also investigate the secondary questions:

- To what extent are the relations between constructed-response SJT scores and other measures of personal, interpersonal, and professional attributes assessed either concurrently or in the future moderated by demographic characteristics (e.g., gender, race, socioeconomic status, language) and methodological variables (e.g., test construction, study design, response instruction)?

To be aligned with modern perspectives on validity, we will frame our study using Messick's framework that emphasizes a uniform conceptualization of validity evidence including content, response process, relations to other variables, internal structure, and consequences, as operationalized in the Standards for Educational and Psychological Testing [24]. In this study, we will synthesize the collected evidence regarding 'relations to other variables'. In this sense, we will use a definition of the validity that includes the relations to both concurrent and subsequent measures.

## Methods

The protocol for this systematic review has been registered in PROSPERO [CRD42022314561].

### Eligibility criteria

Primary studies reporting on the association between scores on constructed-response SJTs with other measures of personal, interpersonal, or professional attributes assessed within the context of training programs for the health professions will be eligible for inclusion. Studies

will be selected according to the following criteria which have also been summarized in Table 1.

**Participants.** We will include studies conducted on students applying to, or training in, a program for the health professions. This could include undergraduate or postgraduate programs in medicine, nursing, dentistry, pharmacology, physical therapy, occupational therapy, chiropractic medicine, veterinary medicine, optometry, nutrition, radiology technology, blood-banking, and laboratory medicine, etc.

**Instruments.** We will include studies reporting on the association of a constructed-response SJT with at least one other measure that, partially or completely, aims to assess personal, interpersonal, or professional attributes. To ensure that the SJT and the outcome measure at least partially assess the same or similar underlying construct, the intended aspects of the evaluation will be recorded. For instance, descriptions from both the SJT and the outcome measure will need to reference at least one of the same aspects of professionalism or situational judgment. These aspects may be components of professionalism definitions or descriptions from health professions governing bodies (e.g., CanMEDS, ACGME) and can include but are not limited to aspects such as interpersonal and communication skills, leadership, reliability, and dependability. If there are no overlapping aspects between the SJT and the outcome measure, the study will be excluded. The measures may be administered at any point throughout the trainee's education to the completion of an SJT. Thus, this will include but not be limited to the Multiple Mini Interview (MMI), Objective Structured Clinical Examination (OSCE), workplace-based evaluations, and personality tests. The purpose of using the SJT may be for selecting applicants to be admitted to a program or for the evaluation of students' performance after admission. We note that we do not require instruments to be the same to look at effects across studies.

**Outcomes.** We will include studies that have reported the magnitude of relation between constructed-response SJT scores and other measures of personal, interpersonal, and professional attributes. These are expected to include univariate and multivariate techniques such as linear and logistic regression analyses, classification techniques, relative risk estimates (e.g., odds ratios) and risk predictions, similar to methods observed in evaluating clinical prediction rules [25]. We also include the relations across demographic characteristics and methodological considerations where available (race, gender, socioeconomic status, language proficiency, level of training, profession).

**Study type.** We will have no restrictions on study design, publication date, or language. Non-empirical literature, letters, commentaries, editorials, meta-analyses, studies without any original data, and reviews will be excluded. We will include unpublished studies as well as published papers to identify as much relevant evidence as possible. While grey literature may

**Table 1. The summary of inclusion and exclusion criteria.**

|  | Inclusion | Exclusion |
|---|---|---|
| Participants | Any persons participating in undergraduate or postgraduate training programs for the health professions | Participants in training programs other than health professions Participants of continuing health professions development or faculty development programs |
| Instruments | Constructed-response SJT Assessment tools that, partially or completely, aim to assess personal, interpersonal, and professional characteristics, either upon admission or in the future | Selected-response SJTs Assessments that mainly measure cognitive domain (GPA, MCAT) |
| Outcome | Validity of constructed-response SJTs Validity of constructed-response SJTs across demographic characteristics | |
| Study type | Published or unpublished primary studies | Non-empirical literature (e.g., letters, commentaries, editorials) and secondary studies (e.g., meta-analysis, reviews) |

cause concerns regarding their methodological quality due to the absence of peer review, inclusion of unpublished data could reduce risk of publication bias, the "file-drawer problem," enrich the power of findings, and reduce research waste [26]. To avoid any potential problem, we will follow the direction of the Cochrane handbook that recommends having at least a similar level of expertise in the review team as a peer reviewer for a journal to appraise unpublished studies [27].

## Information sources

To ensure comprehensiveness of our search, we will use different approaches to identify the relevant studies. To find eligible published studies, we will explore the following electronic databases: MEDLINE (via OVID, 1948 onwards), EMBASE (via OVID, 1980 onwards), CINAHL, ERIC, SCOPUS, and Web of Sciences.

In order to find the grey literature, we will search within OpenGrey, ProQuest Dissertations & Theses, and Electronic Theses Online Service (EThOS). We will also contact academics and researchers and invite them to share raw or summary association information, whitepaper reports, technical quality assurance reports, or conference papers to be included in the study. We will also undertake backward and forward searching by reviewing the reference lists and citations of the included articles to add relevant studies. This will help us identify the works cited in an article and let us find out whether a work has been cited after its publication.

## Search strategy

We will first develop the MEDLINE search strategy. We will focus on two concepts integral to our research question: population (i.e., applicants to, and students in, health professions programs) and instruments of interest (i.e., constructed-response SJTs). We will not include outcomes of interest in our search strategy assuming they would have an impact on the search sensitivity and decrease the number of retrieved articles. The search terms will be identified through reviewing the known relevant papers. These include, for example, BEME guide No. 50 [28] and BEME guide no. 52 [29] for relevant populations of interest, as well as Patterson et al. [6] and Webster et al. [14] for relevant assessments, tests, and/or instruments. Additional terms will be identified using a MeSH analysis grid [30]. The MEDLINE search strategy will be reviewed by a librarian with expertise in systematic review searching and will be revised based on their comments and suggestions. The final MEDLINE search strategy will be modified to adjust the syntax and subject headings of other databases.

A draft search strategy has been included in S1 Table in S1 File.

## Study records

**Data management.** We will use Covidence [31], an online software that supports screening, full-text review, extraction and export of data, to coordinate the study selection process. All retrieved citations and their metadata (i.e., abstract, author names, journal) will be imported into Covidence where first the duplicates will be identified and removed.

**Selection process.** To select studies for inclusion, two reviewers will independently screen the papers in two rounds. The initial screening process will be performed based on the titles and abstracts of all citations via Covidence. For the second round, the full texts of the records will be assessed against the inclusion and exclusion criteria. Studies will be included if both reviewers agree on the relevance. If both reviewers agreed to exclude the paper the article will be rejected. In case of disagreement, the reviewers will resolve the issue by discussion. The potential remaining conflicts will be addressed by a third reviewer. The whole team will meet to discuss and finalize the inclusion of papers.

**Data collection process.**    To extract data from the included studies, we will design a data extraction form which will be outlined in the next section. The form will be revised after pilot testing on two studies. All the extractions will be done independently and in duplicate by two reviewers based on the final data extraction form. If there is any disagreement, first, the coders will discuss the issue and then, if the issue remains unresolved, a third reviewer will independently extract data and then will have a discussion with two coders to reach consensus.

## Data items

We will extract details of the citation, study aim, design, setting, and methodology as well as the assessment tools used and their characteristics. We will also code the key findings and summary notes. If information is not available, it will be indicated as "not reported".

A draft of the data extraction form has been included in S2 Table in S1 File.

## Outcomes and prioritization

### Primary outcomes

- Validity of constructed-response SJTs, operationalized as the correlation between SJT scores and measures of personal, interpersonal, and professional attributes

### Secondary outcomes

- Validity of constructed-response SJTs across demographic characteristics

- Validity of constructed-response SJTs across methodological moderators

### Risk of bias in individual studies

Studies will be assessed for the risk of bias and quality, independently by two reviewers, using the "Quality In Prognosis Studies" tool [32]. This tool is composed of six domains including 31 items which are rated on a four-grade scale. The overall risk is expressed on a three-grade scale (high, moderate, or low) and free-text comments will be used to justify scores. The review team will meet to discuss and resolve any discrepancies until consensus is achieved. The information will be used in the data synthesis and will let us provide a deeper interpretation of the findings. No study will be excluded based on the quality and risk of bias assessment, but all bias and quality outcomes will be reported.

### Data synthesis

We will first provide a description on the characteristics, setting, and context of the included studies. This descriptive synthesis will be used as the basis of synthesis evidence in order to address the review questions. In attempting to answer the review questions, we will synthesize the findings to discuss the validity of constructed-response SJT scores in training programs for the health professions. To improve reporting transparency and mitigate the risk of instrument heterogeneity, a coding scheme will be implemented to organize measures based on their degree of alignment to the underlying construct assessed by the constructed-response SJT. Results from instruments with high overlap, where greater than half the measures score is derived from aspects assessed by the SJT, will be grouped together. Conversely, results from instruments with low overlap will be grouped together. That is, where greater than half the measures score is derived from aspects not assessed in the SJT (e.g., academic, procedural

knowledge or clinical skills). We will test the statistical heterogeneity by calculating the $I^2$ value and examining the width of 80% credibility intervals. If statistical heterogeneity is observed, we will conduct psychometric meta-analyses using a random-effects model. The Pearson correlation coefficients between SJT scores and measures of personal, interpersonal, and professional attributes will serve as the effect size index and will be weighted based on sampling precision (e.g., sample size). All correlation coefficients will be transformed into Fisher's z scale before analyses [33]. We will also report the study findings narratively and will undertake a rich and exploratory descriptive synthesis of evidence to explain the findings.

It is possible that one study may report on several assessment tools or include more than one cohort year of a program. Where unique data for each outcome assessment and participant cohort (year) are available, these will be considered as separate studies.

If adequate data are available, subgroup analyses will be conducted to explore whether different subgroups demonstrate different results. First, all data included in the meta-analysis will be split into subgroups, based on the applicants' characteristics (e.g., gender, race, socioeconomic status, language, profession, level of training) and study characteristics (e.g., study design, publication status). Then, a meta-analysis will be conducted on each of the subgroups. Sensitivity analysis will also be performed in order to explore the impact of different sample size and risk of bias by removing studies that are judged to have small sample size or be at high risk of bias.

## Discussion

The findings of this study will inform best practices for admission and selection of applicants in health professions programs. The association between admission metrics and students' future performance is an important question for medical educators, selection committees, and stakeholders. Determining the validity of constructed-response SJTs and their ability to produce similar inferences across demographic groups can facilitate defensible and scalable holistic admissions practices by programs for the health professions. The findings of this review will also encourage further research of constructed-response SJTs, in particular their validity. We expect this review will identify gaps of knowledge in this field and suggest areas for future research.

## Supporting information

**S1 Checklist. PRISMA-P (Preferred Reporting Items for Systematic review and Meta-Analysis Protocols) 2015 checklist: Recommended items to address in a systematic review protocol.**
(DOCX)

**S1 File.**
(DOCX)

## Acknowledgments

We thank Jacqueline Kreller-Vanderkooy (Learning & Curriculum Support Librarian at the University of Guelph) for her assistance in reviewing and refining our search strategy.

## Author Contributions

**Conceptualization:** Sara Mortaz Hejri, Yoon Soo Park, Haykaz Mangardich, Alexander MacIntosh.

**Investigation:** Xuan Pan, Yoon Soo Park, Amir H. Sam, Haykaz Mangardich.

**Methodology:** Sara Mortaz Hejri, Jordan L. Ho, Xuan Pan, Yoon Soo Park, Amir H. Sam.

**Project administration:** Jordan L. Ho.

**Resources:** Haykaz Mangardich.

**Supervision:** Alexander MacIntosh.

**Writing – original draft:** Sara Mortaz Hejri, Jordan L. Ho, Xuan Pan.

**Writing – review & editing:** Sara Mortaz Hejri, Jordan L. Ho, Yoon Soo Park, Amir H. Sam, Haykaz Mangardich, Alexander MacIntosh.

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
