## [Decision Letter · Decision Letter 0]

21 Jul 2022

PONE-D-22-08111Predictive validity of constructed-response situational judgment tests in health professions education programs: A systematic review and meta-analysis protocolPLOS ONE

Dear Dr. Mortaz Hejri,

Thank you for submitting your manuscript to PLOS ONE. After careful consideration, we feel that it has merit but does not fully meet PLOS ONE’s publication criteria as it currently stands. Therefore, we invite you to submit a revised version of the manuscript that addresses the points raised during the review process.

Two reviewers have assessed the manuscript. They have raised some overlapping concerns about some of the difficulties you may encounter in this analysis, and have provided suggestions for revisions that can be made to the study.

We look forward to receiving your revised manuscript.

Kind regards,

Hanna Landenmark

Staff Editor

PLOS ONE

Journal Requirements:

YSP and AHS have no disclosures to declare. SMH, JLH, XP, and AM disclose they are salaried employees of Altus Assessments which administers a situational judgment test called Casper. The authors receive no reimbursements, fees, or funding related to this study or its outcomes.

3. We note that you have referenced (ie. Bewick et al. [5]) which has currently not yet been accepted for publication. Please remove this from your References and amend this to state in the body of your manuscript: (ie “Bewick et al. [Unpublished]”) as detailed online in our guide for authors

Reviewers' comments:

Reviewer's Responses to Questions

**Comments to the Author**

1. Does the manuscript provide a valid rationale for the proposed study, with clearly identified and justified research questions?

Reviewer #1: Yes

Reviewer #2: Partly

2. Is the protocol technically sound and planned in a manner that will lead to a meaningful outcome and allow testing the stated hypotheses?

Reviewer #1: Partly

Reviewer #2: Partly

3. Is the methodology feasible and described in sufficient detail to allow the work to be replicable?

Reviewer #1: No

Reviewer #2: Yes

4. Have the authors described where all data underlying the findings will be made available when the study is complete?

Reviewer #1: Yes

Reviewer #2: Yes

5. Is the manuscript presented in an intelligible fashion and written in standard English?

Reviewer #1: Yes

Reviewer #2: Yes

6. Review Comments to the Author

You may also provide optional suggestions and comments to authors that they might find helpful in planning their study.

Reviewer #1: I appreciated the clarity and detail in the protocol provided. My primary questions are with regard to feasibility of a meta-analysis on this topic.

1. There are few published empirical studies or dissertations on constructed response SJTs (my quick search identified less than 25) across any setting and even fewer that are in Health Professionals Education programs. Within that, finding multiple studies that look at the same instrument or that look at the same outcome in order to look at cumulative effects seems highly unlikely to me.

2. A second feasibility issue is with regard to the strategy for assessing moderated validity (i.e., differences in predictive validity by subgroup). Such comparisons would require having correlations between predictors and outcomes reported separately by subgroup -- the strategy outlined in the protocol suggests such a split would be possible, but it is rare to see this type of information reported in studies (published or unpublished) because of small Ns. Thus, it does not seem addressing that key question is feasible.

3. It seems that the key predictive validity questions would be focused on outcomes such as performance in classes, graduation rates, etc... Thus, it was unclear to me why looking at a variety of other types of assessments (e.g, personality measures) would be considered an examination of predictive validity rather than convergent validity. Further, because such assessments may be focused on specific constructs that are not those assessed by a given SJT it is unclear what such data will indicate. Perhaps an analysis that focuses on SJTs designed to measure certain attributes to measures designed to measure those same attributes would be a better focus.

4. It was surprising to see the note that SJTs have acceptable internal consistency as a common critique of SJTs are that they are multidimensional at the item level and often do not have high internal consistency.

5. One question regarding constructed responses to assessments are the concerns over how scored -- for example, an oral SJT with constructed responses is akin to a situational interview. It seems looking at dimensions of scoring and who the assessors/raters are would be important aspects to code.

Reviewer #2: I think that meta-analyzing constructed-response SJTs is a worthwhile goal and I would enjoy reading that meta-analysis.

However, I think that to maximize the utility of findings for practitioners, they would need to know more than simply the average predictive validity of constructed-response (C-R) SJTs for selection into Health Professions Education. Based on the broader research on SJTs, we know that this is a method of measurement whose validity can vary greatly as a function of various methodological choices. So I'd expect a practitioner to want to know what methodological choices they should make when designing their C-R SJT in order to maximize its validity. In the case that the average validity is actually not very good, it would be especially important to be able to explain what makes some C-R SJTs more effective than others, rather than the take-home message ending up that these tools are not very useful.

As such, I would recommend that the authors pull all the available studies on C-R SJTs (within Health Professions or even more broadly than that) and determine potential moderators of validity to code and examine in their meta-analysis. The authors can look to the broader SJT research (outside of the Health Professions) for the types of methodological moderators other meta-analyses/reviews have considered (e.g., Campion, Ployhart, & MacKenzie, 2014; Christian, Edwards, & Bradley, 2010; McDaniel, Morgeson, Finnegan, Campion, & Braverman, 2001). While some moderators considered in those studies may not translate to C-R SJTs, others certainly do.

Related to the above, one potential moderator of validity I'd like to highlight is the validity study design (concurrent vs predictive). The authors indicated that they only intend to examine predictive validity designs, but that seems unnecessarily restrictive to me. I don't know about the Health Professions Education field in particular, but from the broader research on SJTs, do know that C-R SJTs constitute a small minority of SJTs in general, and predictive validity studies--as a much less common validity study design than concurrent--further restrict the possible sample of studies to include in a meta-analysis. As a side note, from the authors' description of their plans, it was not entirely clear to me that they definitely plan to only examine predictive validity studies. For example, I did not understand this sentence (lines 137-138): "The measures may be administered at any point throughout the trainee's education to the completion of an SJT." And at the end of that same paragraph, this also doesn't sound like a predictive validity design: "The purpose of using the SJT may be for ... evaluation of students’ performance after admission."

Also, I would expect raters who score responses to C-R SJTs (e.g., their training) to also have validity-moderating potential. I noticed that the example of potential moderators I bring up are actually already reflected in the protocol for what the researchers planned to code about the articles they find (Appendix 2), but the data extraction form may need further refinement to capture anything additional about the research methodology that might be relevant. Again, I believe that the more validity moderators the authors can point to, the more practical utility their research will have.

As a final note, it was unclear to me how the authors plan to examine the effect of possible publication bias (lower quality studies remaining in a file drawer) on their findings (lines 237-238). They may find these sources helpful:

Duval SJ, Tweedie RL. (2000). A non-parametric “trim and fill” method of accounting for publication bias in meta-analysis. Journal of the American Statistical Association, 95, 89–98.

Duval SJ, Tweedie RL. (2000). Trim and fill: A simple funnel plot-based method of testing and adjusting for publication bias in meta-analysis. Biometrics, 56, 276–284.

7. PLOS authors have the option to publish the peer review history of their article (what does this mean?). If published, this will include your full peer review and any attached files.

Reviewer #1: No

Reviewer #2: No

---

## [Author Response · Author response to Decision Letter 0]

19 Sep 2022

Please see the 'Responses to the Reviewers'.

---

## [Decision Letter · Decision Letter 1]

19 Oct 2022

PONE-D-22-08111R1Predictive validity of constructed-response situational judgment tests in health professions education programs: A systematic review and meta-analysis protocolPLOS ONE

Dear Dr. Hejri,

Thank you for submitting your manuscript to PLOS ONE. After careful consideration, we feel that it has merit but does not fully meet PLOS ONE’s publication criteria as it currently stands. Therefore, we invite you to submit a revised version of the manuscript that addresses the points raised during the review process.

We look forward to receiving your revised manuscript.

Kind regards,

Somayeh Delavari, Ph.D.,

Academic Editor

PLOS ONE

Journal Requirements:

Reviewers' comments:

Reviewer's Responses to Questions

**Comments to the Author**

1. Does the manuscript provide a valid rationale for the proposed study, with clearly identified and justified research questions?

Reviewer #1: Yes

Reviewer #2: Yes

Reviewer #3: Yes

2. Is the protocol technically sound and planned in a manner that will lead to a meaningful outcome and allow testing the stated hypotheses?

Reviewer #1: No

Reviewer #2: Partly

Reviewer #3: Partly

3. Is the methodology feasible and described in sufficient detail to allow the work to be replicable?

Reviewer #1: No

Reviewer #2: Yes

Reviewer #3: Yes

4. Have the authors described where all data underlying the findings will be made available when the study is complete?

Reviewer #1: Yes

Reviewer #2: Yes

Reviewer #3: Yes

5. Is the manuscript presented in an intelligible fashion and written in standard English?

Reviewer #1: Yes

Reviewer #2: Yes

Reviewer #3: Yes

6. Review Comments to the Author

You may also provide optional suggestions and comments to authors that they might find helpful in planning their study.

Reviewer #1: While the author(s) did address comments, there was insufficient compelling evidence that there is sufficient literature available for a review

Reviewer #2: I think the revised proposal addresses some of the issues with the original, but there are still a few points that could use clarification.

1. In using Messick's validity framework, I think the authors should also adjust their terminology and refer to the test's "validity" or "construct validity," as opposed to its "predictive validity," which has a narrower definition and can be confusing to readers as referring to study designs where the SJT predicts an outcome measure separated in time from the SJT (predictor).

2. I don't think the other reviewer's concern has been adequately addressed regarding the authors being able to find multiple studies that look at the same instrument or that look at the same outcome in order to be able to consider cumulative effects. The authors say this is not a requirement for them to be able to do their analyses, but I did not see a rationale in the response to reviewers or the revised proposal. It's unclear to me what the analyses would indicate without attending to the constructs assessed by the instruments involved.

3. While the authors acknowledge in their responses that it is important to code potential moderators of validity in their review of the literature, there is no research question posed in the proposal intro to look at methodological moderators of validity as part of their analyses.

Reviewer #3: The authors are planning to assess the predictive validity of constructed response situational judgement tests for professional performance in the healthcare setting in an upcoming systematic review, the protocol for which is presented here. The issue of the validity of university enrollment tests, especially those that assess humanistic tendencies is a concerning one and the question has been formulated well. The protocol has also been written very well with adequate methodological details. There are only a few remarks and suggestions that the authors may wish to consider before publication of the protocol and conduction of the systematic review.

1. The term "health professions education programs" was a bit confusing to me at the beginning and made me think of programs that aim to train educators rather than health professionals. Thus, I suggest that the authors replace it with another more clear term like health sciences/professional training programs.

2. The outcomes and the acceptable assessment tools that assess “other measures of personal, interpersonal, and professional attributes” warrant further discussions and clarifications because these are the tests against which your main test is being tested and needs to be validated. Do the authors have pre-specified inclusion criteria for these tests? Is any comparison considered acceptable? I assume this issue can impact the validity of the final results because these measures will vary in their discriminative and diagnostic features. I think in this case, your study can also be formulated into a diagnostic question where these measures may represent the “reference standard”. The same case goes for clinical prediction rules which are similar to this study.

3. While the currently suggested risk of bias tool seems appropriate and meets the general standards, I suggest that the authors also consider the PROBAST (Prediction model Risk Of Bias Assessment Tool) which may be more helpful for their question which is a mixture of diagnosis and prognosis.

7. PLOS authors have the option to publish the peer review history of their article (what does this mean?). If published, this will include your full peer review and any attached files.

Reviewer #1: No

Reviewer #2: No

Reviewer #3: No

---

## [Author Response · Author response to Decision Letter 1]

5 Dec 2022

Please see the attached 'Response to Reviewers' document for a point-by-point response with the associated clean and tracked changes versions of the manuscript.

---

## [Decision Letter · Decision Letter 2]

2 Jan 2023

Validity of constructed-response situational judgment tests in training programs for the health professions: A systematic review and meta-analysis protocol

PONE-D-22-08111R2

Dear Dr. MacIntosh,

We’re pleased to inform you that your manuscript has been judged scientifically suitable for publication and will be formally accepted for publication once it meets all outstanding technical requirements.

Kind regards,

Somayeh Delavari, Ph.D.,

Academic Editor

PLOS ONE

Additional Editor Comments (optional):

Reviewers' comments:

Reviewer's Responses to Questions

**Comments to the Author**

1. Does the manuscript provide a valid rationale for the proposed study, with clearly identified and justified research questions?

Reviewer #2: Yes

Reviewer #3: Yes

Reviewer #4: Yes

2. Is the protocol technically sound and planned in a manner that will lead to a meaningful outcome and allow testing the stated hypotheses?

Reviewer #2: Yes

Reviewer #3: Yes

Reviewer #4: Yes

3. Is the methodology feasible and described in sufficient detail to allow the work to be replicable?

Reviewer #2: Yes

Reviewer #3: Yes

Reviewer #4: Yes

4. Have the authors described where all data underlying the findings will be made available when the study is complete?

Reviewer #2: Yes

Reviewer #3: Yes

Reviewer #4: Yes

5. Is the manuscript presented in an intelligible fashion and written in standard English?

Reviewer #2: Yes

Reviewer #3: Yes

Reviewer #4: Yes

6. Review Comments to the Author

You may also provide optional suggestions and comments to authors that they might find helpful in planning their study.

Reviewer #2: I think the authors have addressed the reviewers' comments. I don't have further feedback at this time.

Reviewer #3: Good luck to the authors on conducting their perfectly designed review. Thank you for responding to my concerns with details.

Reviewer #4: thank you for choosing such an interesting topic. It is needed to add the search syntax at least for one database with its NNR.

7. PLOS authors have the option to publish the peer review history of their article (what does this mean?). If published, this will include your full peer review and any attached files.

Reviewer #2: No

Reviewer #3: No

Reviewer #4: **Yes: **Manoosh Mehrabi

---

## [Editor Report · Acceptance letter]

18 Jan 2023

PONE-D-22-08111R2 

Validity of constructed-response situational judgment tests in training programs for the health professions: A systematic review and meta-analysis protocol 

Dear Dr. MacIntosh:

I'm pleased to inform you that your manuscript has been deemed suitable for publication in PLOS ONE. Congratulations! Your manuscript is now with our production department. 

Kind regards, 

on behalf of

Dr. Somayeh Delavari 

Academic Editor

PLOS ONE